# Biomechanical–Structural Correlation of *Chordae tendineae* in Animal Models: A Pilot Study

**DOI:** 10.3390/ani11061678

**Published:** 2021-06-04

**Authors:** Justyn Gach, Izabela Janus, Agnieszka Mackiewicz, Tomasz Klekiel, Agnieszka Noszczyk-Nowak

**Affiliations:** 1Department of Internal Medicine and Clinic of Diseases of Horses, Dogs and Cats, Faculty of Veterinary Medicine, Wrocław University of Environmental and Life Sciences, Grunwaldzki sq. 47, 50-366 Wrocław, Poland; jjgach@gmail.com; 2Department of Pathology, Faculty of Veterinary Medicine, Wrocław University of Environmental and Life Sciences, C. K. Norwida 31 Street, 50-376 Wrocław, Poland; izabela.janus@upwr.edu.pl; 3Department of Biomedical Engineering, University of Zielona Góra, Prof. Z Szafrana 4 Street, 65-516 Zielona Góra, Poland; a.mackiewicz@iimb.uz.zgora.pl (A.M.); t.klekiel@ibem.uz.zgora.pl (T.K.)

**Keywords:** *Chordae tendineae*, heart, canine, swine, biomechanics, histopathology

## Abstract

**Simple Summary:**

The *Chordae tendineae* are part of the atrioventricular apparatus. They are mainly responsible for the mechanical functions of heart valves. Degenerative mitral valve disease is the most common heart disease in dogs and is responsible for about 75% of cases of heart failure. One of the complications of this disease is *Chordae tendineae* rupture. It is clinically relevant to better understand the biomechanical and structural properties of CT in order to begin further studies about biomarkers suggesting an episode of CT rupture. Such an episode leads to acute pulmonary oedema and worsens the clinical status of the patient. Information about the biomechanical and structural properties of healthy CT and CT affected by the degenerative process are essential in understanding how CT behave in an in vivo environment.

**Abstract:**

The mitral valve apparatus is a complex structure consisting of the mitral ring, valve leaflets, papillary muscles and *Chordae tendineae* (CT). The latter are mainly responsible for the mechanical functions of the valve. Our study included investigations of the biomechanical and structural properties of CT collected from canine and porcine hearts, as there are no studies about these properties of canine CT. We performed a static uniaxial tensile test on CT samples and a histopathological analysis in order to examine their microstructure. The results were analyzed to clarify whether the changes in mechanical persistence of *Chordae tendineae* are combined with the alterations in their structure. This study offers clinical insight for future research, allowing for an understanding of the process of *Chordae tendineae* rupture that happens during degenerative mitral valve disease—the most common heart disease in dogs.

## 1. Introduction

The left atrioventricular apparatus is located between the left atrium and the left ventricle and consists of the following: mitral ring, anterior and posterior cusps, *Chordae tendineae*, and papillary muscles [1]. *Chordae tendineae* (CT) are columnar structures found only in mitral and tricuspid valves. CT are divided into three groups: first-order CT, which connect the leaflets and the papillary muscles, second-order CT located between the midventricular surface of both cusps and the papillary muscles, and third-order CT, which extend between the parietal leaflet and the ventricular wall [2,3]. The CT modulate the transmission of strain from the papillary muscles to the valve leaflets, as evidenced by the heterogeneity of the mechanical properties and structure of the chordal areas of adhesion on the leaflet [4]. The area of the valve leaflet without CT insertion regions is circumferentially stiffer than the area with CT attachments. In normal CT, the orientation of collagen fibres is complex throughout the area of the insert. This enables the three-dimensional transmission of strain through the CT [5].

Myxomatous mitral valve disease (MMVD) is the most common acquired cardiovascular disease in dogs, and accounts for approximately 75% of cases of chronic heart failure [6,7,8,9]. The etiology of the myxomatous process is still unknown [10]. Myxomatous mitral valves are characterized by a disorganization of the structural elements of the leaflets and a weakening of the *Chordae tendineae* as well. These changes cause a significant loss of the valve’s mechanical properties, and often lead to valve prolapse and/or mitral regurgitation. In myxomatous mitral leaflets, collagen and glycosaminoglycan accumulation is observed [5]. This degenerative process also affects the CT [10]. The American College of Veterinary Internal Medicine (ACVIM) distinguishes four stages of MMVD. Stage A includes breed-predisposed dogs (e.g., Cavalier Kings Charles Spaniel, Yorkshire terrier, Maltese, Chihuahua, Dachshund) at high risk for developing heart failure in the future. Such patients show no clinical signs and no structural changes in the heart during the echocardiography examination. Stage B1 includes dogs without clinical signs but with a heart murmur present due to mitral regurgitation secondary to mitral valve leaflet degeneration, without enlargement of the heart chambers. Stage B2 includes dogs with structural heart disease (chamber enlargement). Stage C includes patients with current left ventricular heart failure symptoms or a history of such symptoms (exercise intolerance, cough, tachypnoe, dyspnea). Patients who have developed heart failure and present with recurrent heart failure symptoms despite standard treatment are classified as stage D. In stages C and D, a number of complications such as *Chordae tendineae* rupture, development of pulmonary hypertension and left atrial tear can occur [11].

The prevalence and progression of MMVD is strongly associated with age, breed and gender [8,12,13]. It occurs more often in small breeds than in large breeds. Males seem to be more vulnerable than females and can develop MMVD earlier [14,15]. The disease usually progresses over many years and morbidity is dependent on valve regurgitation and volume overload [16]. MMVD is recognized during auscultation of a heart murmur. A left apical holosystolic murmur is typical of mitral regurgitation [17]. However, definitive diagnosis is made by performing an echocardiographic exam. During this examination, it is possible to see changes in the valves (nodular and thickened appearance), enlargement of the heart chambers, and valve regurgitation [11].

The changes in leaflet composition and mechanics due to progressive degeneration are already known but, to the authors’ knowledge, histological and mechanical changes in CT have not yet been evaluated in dogs. In our studies, we examined the biomechanical and structural characterisation of canine and porcine CT and the correlation between those properties. We evaluated the hypothesis that the histological and structural remodelling of CT (in addition to the valve leaflets) has an influence on the mechanical characteristics of CT and can contribute to a higher probability of CT rupture.

Our in vitro results will provide a further insight into the biomechanical characteristics of *Chordae tendineae* and their functionality in vivo. This will allow us to start research on biomarkers that anticipate an episode of *Chordae tendineae* rupture in the course of degenerative mitral valve disease. This has clinical relevance because CT rupture leads to acute pulmonary oedema and worsens the patient’s prognosis and general condition. Our study comprised two examinations: biomechanical and histopathological. The biomechanical examination of CT samples was based on static uniaxial tensile tests. Such tests have been used on porcine, ovine and human but not on canine CT [18,19,20].

The connective tissue that structures the *Chordae tendineae* consists of collagen and elastic fibres. The first are largely responsible for the stiffness of the tissue, while the second are responsible for its elastic properties. The accurate proportion and arrangement of the collagen and elastic fibres guarantees the CT’s functional properties [21,22]. The tissue structure, including the orientation of collagen fibres, affects the stress–strain relation. In addition, the structure of the collagen fibres causes the anisotropic mechanical properties observed during degenerative changes in the *Chordae tendineae*, and thus affects the regurgitation of the valves [23]. The structure of the CT enables them to be highly deformable in the axial direction during the cyclic operation of the heart valves. Knowledge of the mechanical parameters is important for the proper assessment of the condition and functionality of the *Chordae tendineae*, as well as for the development of new techniques for the reconstruction of broken CT [24,25].

## 2. Materials and Methods

The material for this study consisted of *Chordae tendineae* collected post-mortem from the hearts of dogs (n = 27) and domestic pigs (n = 11). From each dog, we obtained 5–7 CT; from each pig, we obtained 5 or 6 CT. From each animal, two *Chordae tendineae* underwent histopathological examination and the rest underwent biomechanical examination. Pigs were 14 weeks old and dogs’ ages varied from 8 to 15 years. Animals were represented by both males and females (Table 1). Porcine *Chordae tendineae* served as material for the preliminary study in which the research methodology was developed. They also served as a control group, as the CT were healthy and not affected by the degenerative process. After the preliminary study, a corresponding study was performed on canine CT.

After collecting each heart, the mitral valve was dissected and rinsed with Biolasol solution (Biochefa, Sosnowiec, Poland), and CT were collected for further analysis. The Biolasol solution preserves the tissue and does not change its mechanical properties [26,27,28,29]. At least two CT from each valve leaflet were collected for histopathological examination and fixed with 7% buffered formalin solution for 24 h. The tissues were fixed at room temperature. The remaining CT were stored at a reduced temperature (4 °C) in sterile containers filled with Biolasol solution (Biochefa, Sosnowiec, Poland) for the biomechemical examination. The composition of Biolasol slows down the negative processes taking place in organs and stabilizes them. The method employed in the dissection and traction of *Chordae tendineae* was designed to avoid, to the maximum possible extent, the degeneration of the material under investigation. Every chorda tendinea was measured in length, diameter and area before biomechanical examination; length was measured after examination as well (Table 1).

CT underwent two examinations: a biomechanical and a histopathological one.

### 2.1. The Biomechanical Examination

The non-linear and anisotropic response of *Chordae tendineae* is time-dependent; therefore, the time between euthanasia of the animal and completion of the mechanical strength test was reduced to a minimum. In this examination, the time from the moment of euthanasia to the end of the strength test did not exceed 24 h.

In order to determine the strength characteristics of *Chordae tendineae*, a static uniaxial tensile test was performed on each of the samples from pigs and dogs [19]. During the investigation, a velocity of 5 mm/min, used in the literature during soft tissue testing, was applied. For static tensile testing of *Chordae tendineae*, a Zwick/Roell EPZ 005 testing machine (ZwickRoell GmbH & Co. KG, Ulm, Germany) was used (Figure 1A). The *Chordae tendineae* were stretched in an axial direction (along the long axis of the chorda tendinea) while fastened into a manufactured stand dedicated to stretching the mentioned specimens (Figure 1B). *Chordae tendineae* were dissected with a minimum of papillary muscle and valve leaflet. Every CT was sandwiched with sandpaper on both ends (to make sure there was no slip between them) and fixed into the grips (Figure 1B,C). This method made it possible to stretch only the CT, not the elements connected to it (papillary muscle and valve cusps). The static uniaxial tensile test was performed until the CT ruptured. Before testing, the samples were parameterized using digital images and Zeiss AxioVision Rel software version 4.8; thus, exact values of the *Chordae tendineae*’s dimensions were obtained, i.e., cross-sectional area and length (Figure 1C) [30].

This method of determining the geometric parameters of the samples was chosen due to the fact that the soft tissues could be damaged by using another manual measurement instrument, which could therefore result in incorrect mechanical characteristics. Force-displacement characteristics from tensile tests were used as the basis for further analysis. The knowledge of the relationship between the cross-sectional area and longitudinal dimensions of the specimens allowed us to convert the obtained data into stress and strain values in Microsoft Excel.

During our studies, we decided to additionally investigate a chorda tendinea from the tricuspid valve from one dog’s heart and compare it to the results obtained from CT of the mitral valve in that dog.

### 2.2. The Histopathological Examination

After the 24-h fixation in the 7% buffered formalin solution, the samples were embedded in paraffin blocks, sectioned at 6 μm and placed on microscopic slides. The slides were stained using the standard haematoxylin–eosin method, Masson–Goldner trichrome staining, and elastic red–picrosirius staining for better visualisation of the collagen and elastic fibres. The slides were evaluated using a Leica DM500 microscope coupled to a Leica ICC50W camera and LAS Interactive Measurement computer software (LAS Interactive Measurement, Leica, Germany). The composition and layout of connective tissue fibres were evaluated.

The mitral valves were assessed according to Whitney grading [31]. Valves graded 0 were considered normal and grades I–IV were considered MMVD.

The statistical analysis was performed using the STATISTICA package (data analysis software system), version 13 by TIBCO Software Inc. (2017). An independent samples *t*-test was used; all data were displayed as the mean ± standard deviation and statistical significance was set at 5%. 

## 3. Results

### 3.1. The Biomechanical Examination

All samples that underwent biomechanical testing are shown in Table 1.

During the biomechanical examination force–elongation characteristics were obtained and converted into strain and stress (Figure 2), determining the basic mechanical parameters such as tensile strength and Young’s modulus.

The obtained characteristics showed a strong non-linear character of the tensile response of *Chordae tendineae*. The shape of the curve showed a hyper-elastic model of this soft tissue material. It is worth mentioning that the curves were heterogeneous and showed a loss of stability when certain values of tensile force were exceeded. This shows the complex structure of the string and the arrangement of the fibres in bundles. When the strength of individual fibres was exceeded, there was a momentary decrease in strength, so the curve showed a decrease and then an increase to the strength of the next stretched fibre, etc. The maximum strength of a chorda tendinea is described by the breaking point of the strongest fibre. At the same time, the stretch diagrams showed micro-creeping, characteristic of soft tissues, which should be explained in more detail, together with the determination of the exact mathematical model of the tissue endurance of *Chordae tendineae*. This creep is related to the structure of the connective tissue that builds up the CT, which consists of elastin and collagen fibers that change their configuration inside the matrix during load application and removal.

The samples were tested at random, hence the large dispersion in the obtained values of tensile strength. In the case of pigs, the tensile strength had values of 3.19–12.419 MPa (average 7.522 ± 3.6 MPa). In the case of *Chordae tendineae* from healthy dogs, the tensile strength varied from 4.61 to 23 MPa (average 10.86 ± 5.60 MPa). Significantly lower strength values were observed for *Chordae tendineae* affected by the degenerative process. The highest value of the obtained tensile strength in these samples did not exceed 5.5 MPa and the determined average was 3.36 ± 1.88 MPa. Statistically significant differences were observed between healthy and degenerated CT (*p* = 0.022). Comparison of the CT from healthy dogs and pigs revealed no statistically significant differences between them (*p* = 0.15). Examples of strain–stress ratio curves for a healthy dog and a dog with degenerated *Chordae tendineae* are shown in Figure 3.

### 3.2. Comparison of CT from Mitral and Tricuspid Valve

The average tensile strength value of the CT from the mitral valve was 13.96 ± 9.66 MPa, while that of the CT from the tricuspid valve was 3.6 ± 3.67 MPa. The differences are shown in Figure 4.

In order to determine the strength values of the *Chordae tendineae*, nonlinear stress–strain relationships for plot models and polynomial regression plots were developed at a parameter level of R2 ≥ 86.3. The strength values take elastic standard errors into account. The measurement error of the testing machine was ±0.04 N, while the linear measurements from the digital images were accurate to 1 pixel.

### 3.3. The Histopathological Examination

The histopathological examination of the CT showed differences in structure between animals with normal valves and those suffering from mitral valve disease. The *Chordae tendineae* obtained from pigs and dogs with normal mitral valves showed a regular structure. The fibres were arranged collaterally, and the CT were regular in shape and diameter. In contrast, during mitral valve disease, the layout of connective tissue fibres became chaotic and disarranged. The elastic fibres showed segmental absence in the affected CT. Moreover, the *Chordae tendineae* from animals suffering from mitral valve disease showed segmental thickening (Figure 5 and Figure 6).

We also performed a histopathological examination of a ruptured CT. The structure was completely torn, with free spaces between the fibre bundles visible (Figure 7 and Figure 8).

## 4. Discussion

Myxomatous mitral valve disease is the most common form of valvular heart disease in dogs. Mitral valve closure depends on the proper functioning of each component of the valve apparatus, including the *Chordae tendineae* [32]. Any change in the geometry and topology of *Chordae tendineae* distribution in the valve apparatus affects the mechanics of mitral valve function [33].

Many studies of *Chordae tendineae* and their structural and biomechanical features have been already conducted, but there is still a lack of studies about this topic in dogs. The histopathological examination of *Chordae tendineae* allowed us to evaluate their structure, while the biomechanical study allowed us to evaluate their mechanical properties. Analysis of these two properties allowed us to confirm our hypothesis. Our study showed that changes in the mechanical persistence of *Chordae tendineae* are combined with changes in their structure. *Chordae tendineae* affected by the degenerative process showed lower mechanical strength and ruptured under lower force during the uniaxial tensile test.

Although the changes in mitral valve leaflets in MMVD are well described in the literature, information regarding CT structural remodelling is lacking. In a paper describing a porcine model of CT in both mitral and tricuspid valves, Pokutta-Paskaleva et al. [18] note a relationship between the distribution and presence of interfibrillar gaps in the elastic fibres and the extensibility of the CT. In addition, the pattern of collagen fibres in human and ovine CT was linked to variations in rupture properties [19]. Moreover, the variations in the composition of CT depending on their location in the valve apparatus were linked to their different stress relaxation values. This shows the role of not only the arrangement of collagen and elastic fibres, but also the total amounts of these fibres and glycosaminoglycans in the process of CT elongation and rupture [23]. The literature describes models of healthy mitral and tricuspid valves. Therefore, it is highly possible that the histopathological remodelling of CT during MMVD can contribute to changes in *Chordae tendineae* strength values and should be further investigated in dogs with different stages of MMVD.

### 4.1. The Way Forward

There is a lack of information about tricuspid valve CT biomechanical properties [4]. In our study, we examined *Chordae tendineae* from the tricuspid valve apparatus, and found significant differences (as indicated in the Results section). There is still a need for further study of the tricuspid valve CT and its biomechanical features with a larger sample size.

In the case of *Chordae tendineae*, as reported in the literature, special attention should be paid to the position of the chordae in the valve apparatus, as their structure differs. In one study, a difference in tension was observed between CTs attached to the anterior papillary muscle (APM) and to the posterior papillary muscle (PPM) [28]. An examination of particular types of CT dependent upon their location in the heart (grouped by leaflet type or chordal insertion, i.e., first, second and third order CT) should be performed in order to link strength values to histological and microstructural status [19,20,21,22,34,35,36,37]. That may be a cause of the large dispersion of results observed in our studies [23,38].

Furthermore, we observed that differences in *Chordae tendineae* strength values were not related to the CT length. This can explain the fact that strength (and thus the chance of a CT rupture episode) is not dependent on the size of the heart and its structures, or the size of the organism. Further studies should include such information in their Results section to confirm this hypothesis. 

Moreover, detailed histopathological investigations are required, e.g., an immunohistochemical study. It is clinically relevant to investigate the effect of treatment of the primary disease (degenerative valve disease) on the biomechanical properties of CT [5]. This will allow us to study the response to treatment at the cellular and biomechanical levels.

### 4.2. Limitations

There are some limitations to this study. First, there was a lack of clinical information from some dogs, and no follow-up period for the disease progression in them. Second, it was not possible to assess the correlation of MMVD-induced changes with dog breeds. Moreover, no statistical analysis was performed during the comparison of mitral and tricuspid valve CT because the groups were too small to perform such an analysis. Thus, further studies should include larger groups. The last limitation was that some CT were too short, so it was not possible to clamp them into claws, and therefore they had to be removed from statistical analysis.

## 5. Conclusions

This study aimed to evaluate the correlation between the biomechanical and histopathological properties of the *Chordae tendineae* on animal models. Healthy CT and CT altered due to degenerative processes underwent both biomechanical and histopathological examination. CT from dogs and pigs were compared as well. Healthy CT showed a regular microstructure, regular shape and diameter, while the structure of degenerated CT was chaotic and disarranged. Moreover, the CT from animals suffering from degenerative disease showed segmental thickening. Significant differences were found in tensile strength; healthy CT had higher values than altered CT. Further studies based on this pilot study may be carried out.

## Figures and Tables

**Figure 1 animals-11-01678-f001:**
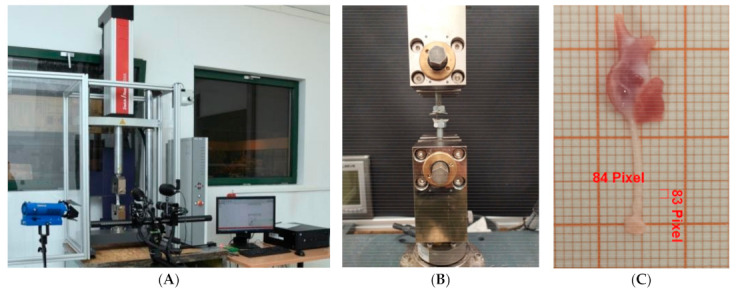
Static tensile test: (**A**)—Zwick/ROell EPZ 005 testing machine, (**B**)—the specimen clamping, (**C**)—a digital specimen parameterization; papillary muscle is at the top and valve part is at the bottom.

**Figure 2 animals-11-01678-f002:**
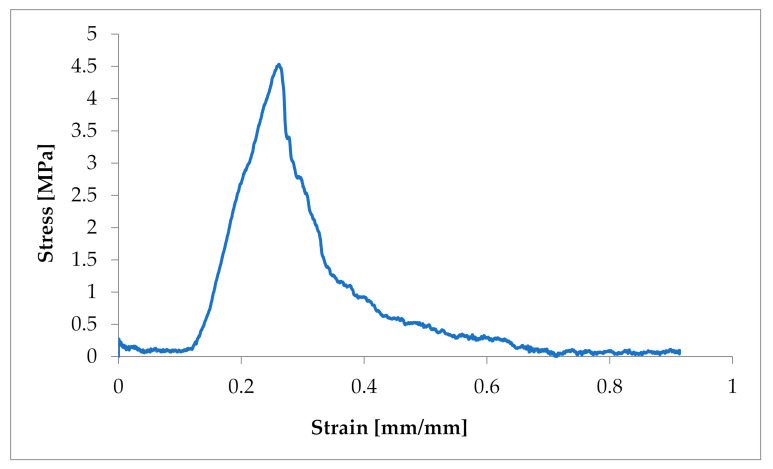
Example of a strain–stress ratio curve in one CT from a dog suffering from MMVD.

**Figure 3 animals-11-01678-f003:**
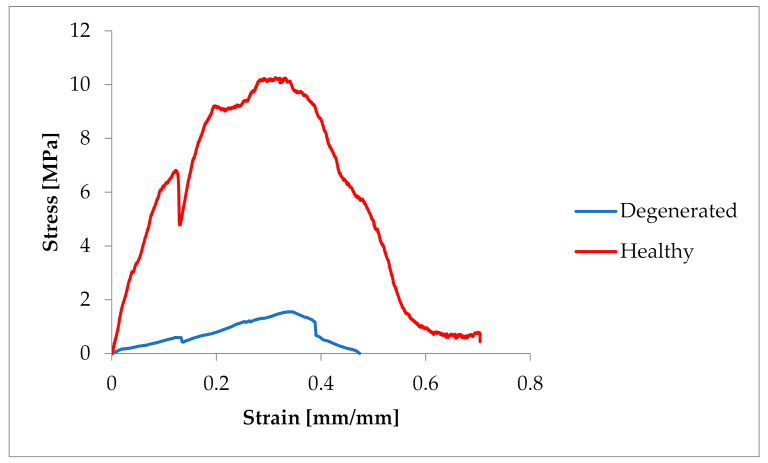
Comparison of the strain–stress ratio curve in healthy and degenerated CT.

**Figure 4 animals-11-01678-f004:**
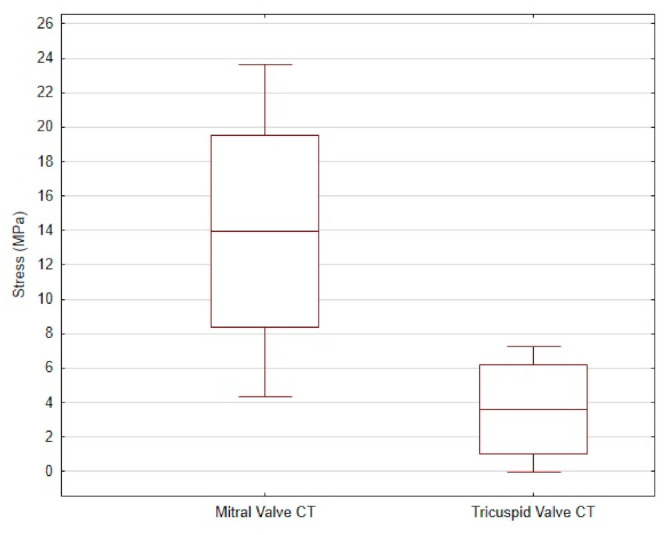
The mean and standard deviation of the stress values in the mitral valve CT and the tricuspid valve CT of dogs.

**Figure 5 animals-11-01678-f005:**
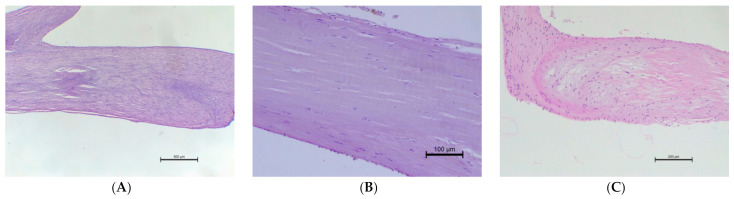
The microscopic examination of *Chordae tendineae* from normal and diseased mitral valves. (**A**) Normal chorda tendinea from pig; HE stain; magnification 40×. (**B**) Normal chorda tendinea from a dog; HE stain; magnification 200×. (**C**) Altered chorda tendinea from dog suffering from mitral valve disease—disarrangement in the filaments is visible; HE stain; magnification 100×. The left side of each figure is the attachment point to the valve, the right side is that to the papillary muscle.

**Figure 6 animals-11-01678-f006:**
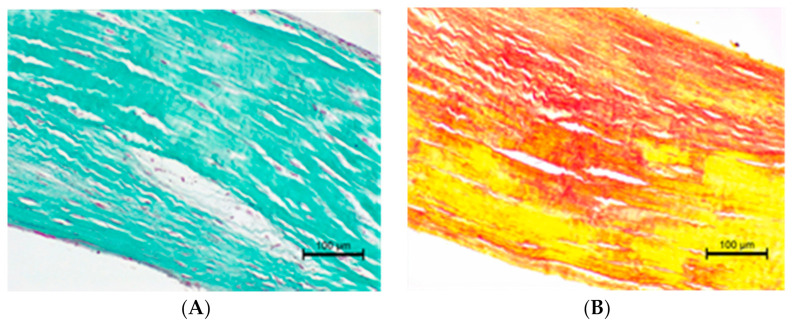
(**A**) Altered chorda tendinea from dog suffering from mitral valve disesase—disarrangement in collagen filaments is visible; Masson–Goldner trichrome stain; magnification 200×. (**B**) Altered chorda tendinea from dog suffering from mitral valve disesase—disarrangement and absence of elastic filaments in some parts of the tissue is visible; elastic red–picrosirius stain; magnification 200×. The orientation is the same as in Figure 5.

**Figure 7 animals-11-01678-f007:**
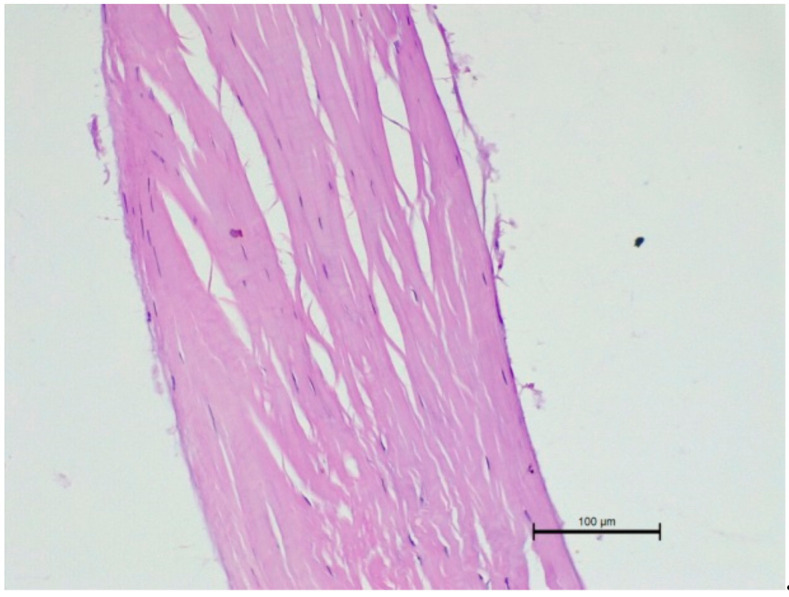
Microscopic examination of a ruptured chorda tendinea from a dog. A severe disarrangement of filaments is visible. HE stain; magnification 200×. Valve side is at the bottom of Figure, while the papillary muscle part is at the top.

**Figure 8 animals-11-01678-f008:**
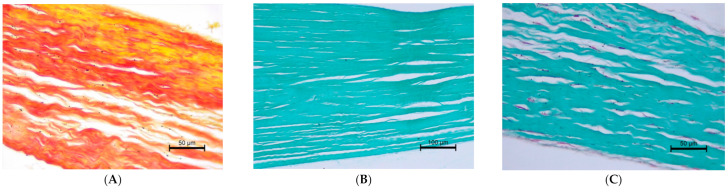
Ruptured *Chordae tendineae*. (**A**) elastic red–picrosirius stain; magnification 400×. (**B**) (magnification 200×) and (**C**) (400×) Masson–Goldner trichrome stain. The valve part is at the left and papillary muscle part at the right of each Figure.

**Table 1 animals-11-01678-t001:** Characteristics of each CT taken for biomechanical examination.

Animal Model	Sex	CT No.	Type of Valve	Before Biomechanical Test			Post Biomechanical Test
				Length [mm]	Diameter [mm]	Surface Area [mm^2^]	Length [mm]
Dog 1	Male	1	Mitral	8.638	0.294	0.271	16.926
		2	Mitral	6.584	0.562	0.991	16.254
		3	Mitral	8.218	0.445	0.623	16.708
		4	Tricuspid	9.551	0.402	0.507	11.422
		5	Tricuspid	9.748	0.271	0.230	11.934
Dog 2	Female	1	Mitral	9.577	0.421	0.557	13.126
		2	Mitral	7.489	0.367	0.422	12.222
		3	Mitral	7.488	0.310	0.302	11.976
Dog 3	Male	1	Mitral	8.306	0.382	0.458	15.894
		2	Mitral	8.093	0.252	0.199	18.290
		3	Mitral	9.360	0.984	3.039	16.875
		4	Mitral	8.309	0.382	0.458	21.193
Dog 4	Male	1	Mitral	25.458	0.768	1.852	18.255
		2	Mitral	18.810	0.343	0.37	16.049
		3	Mitral	14.072	0.369	0.428	19.087
Dog 5	Male	1	Mitral	10.078	0.505	0.800	16.994
		2	Mitral	10.370	0.758	1.803	15.966
Pig 1	Female	1	Mitral	10.013	0.704	1.557	20.289
		2	Mitral	15.746	0.772	1.872	22.760
		3	Mitral	8.935	0.804	2.030	21.865
Pig 2	Female	1	Mitral	10.725	0.805	2.033	24.245
		2	Mitral	8.246	0.762	1.823	12.582
		3	Mitral	8.735	0.657	1.355	17.768
		4	Mitral	10.947	0.540	0.915	15.725

## Data Availability

The data presented in this study are available on request from the corresponding author.

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
