# Peer review of "Biomechanical–Structural Correlation of Chordae tendineae in Animal Models: A Pilot Study"

_animals, 2021, doi:10.3390/ani11061678_

Round 1

Reviewer 1 Report

This work is an interesting analysis on the correlation between biomechanical and histopathological properties of the chordae tendineae on animal models.

Generical comments/observations:

  • Interesting could be to show the results for a different type of CT (i.e., basal, marginal).
  • In figure 3: Comparison of the strain-stress ratio curve in healthy and degenerated CT, how do you explain the reversal peak observed in the healthy Stress-strain curve (red) around the strain 0.1?
  • Figure 4 is not cited in the text;
  • Related to this figure 4, Could be an interesting comment and discussion, the not negligible variability observed in terms of strain for the Mitral Valve CT. It is possible to correlate the behavior with the pathology type?
  • Line 140: "Chordae tendineae were dissected with only a small part of papillary muscle and valve leaflet": This small should have been quantified from a mechanical point of view;
  • Figure 5 could be more impressive if they had not used a healthy pig so compare with an unhealthy dog; Should have been the same animal from my point of view;
  • Other than the above, perfect conclusion and structure.

Author Response

Thank you for all your comments and feedback. I attach the corrected version. 

May I answer for some questions here?

,,Interesting could be to show the results for a different type of CT (i.e., basal, marginal).''

This is something we wrote in ,,The way forward''. As it was a pilot study (we are changing the title adding ,,pilot study'' as it was recommended from 2nd reviewer) we didn't examine each type of chordae tendineae. We will do it in next studies.

,,In figure 3: Comparison of the strain-stress ratio curve in healthy and degenerated CT, how do you explain the reversal peak observed in the healthy Stress-strain curve (red) around the strain 0.1?''

The structure of CT is complex, during stretching some of the individual fibres break, so that the whole structure loses stability for a while, which can be seen on the diagram as a reversal peak. During further stretching, the remaining fibres take up the force, stretching further until all the fibres break (moment of CT rupture).

,,Related to this figure 4, Could be an interesting comment and discussion, the not negligible variability observed in terms of strain for the Mitral Valve CT. It is possible to correlate the behavior with the pathology type?''

In latest version there was a mistake, wrong Figure was attatched. We corrected it, so instead of Strain there are Stress values. 

,,Line 140: "Chordae tendineae were dissected with only a small part of papillary muscle and valve leaflet": This small should have been quantified from a mechanical point of view''

The muscle was placed in the jaws of the testing machine, and only CT was stretched during static uniaxal tensile test. The muscle did not affect the mechanical characteristics of the CT. 

The other corrections can be found in the Word doc.

Once again thank you for your work.

Best regards,

Justyn Gach

Reviewer 2 Report

Gach et al. examined the biomechanical and structural properties of chordae tendineae collected from canine and porcine hearts in vitro.  

The following points must be addressed:

  1. The title is “Biomechanical structural correlation of chordae tendineae in animal models”. However, this paper contents are preliminary. Please consider adding the title 'a pilot study' or 'limitation study'.

  1. How did you distinguish between healthy dogs and dogs with MMVD, between normal CT and CT with MMVD, degenerated CT (Line 202, Line 207, Lines 209-210, Line 242)?  Please explain the reason.

  1. Show the table including the length, diameter, area, tensile strength value for each CT samples obtained from the MV and TV, before and after the biomechanical examination.

  1. Show the table including the number of CT samples obtained from each mitral valve and tricuspid valve?

5.Need more histopathological and immunohistological analyses of the CT samples after the biomechanical examinations.

Minor comments:

Line 14.  Add the abbreviation:   Chordae tendineae (CT)

Materials and Methods

Line 111.  Dog’s age varied from 8 to 15.  Is the age ‘years old’ or ‘weeks old’?

Lines 118-120. This study used at least two CT from each valve leaflet for histopathological examination. The remaining CT were stored in Biolasol solution for the biomechanical examination.  Therefore, describe the total number of CT samples obtained and the number of CT samples used for the biomechanical experiment.

Line 120. Fixed with 7% buffered formalin solution for 24 hours. Is 7% the correct concentration?  Describe the temperature for fix.

Line 168.  Add the abbreviation:   standard haematoxylin-eosin (HE)

Figure 1, 2, 3, 4, 5, 6, and 7 should be annotated.

For example,

Figure 1(C) Describe the orientation. Which is the papillary muscle or valve cusps?

Figure 2. Is this data from pig?  Describe the number of the CT samples. Is this data healthy CT sample as a control?

Figure 3. Is this data from dogs? Describe the number of the CT samples used for this study.

Figure 4. Is this data from dogs?  Describe the number of the CT samples obtained from MV and TV.

Figure5, 6, and 7.  Describe the orientation. Which side is the valve attached to?

Author Response

Thank you for all your comments and feedback. I attach the corrected version.

Best regards,

Justyn Gach
